# Influence of analysis conditions for antimicrobial susceptibility test data on susceptibility rates

Yasutoshi Hatsuda[1]*, Toshihiko Ishizaka[2], Naonori Koizumi[2], Yukako Yasui[2], Takako Saito[2], Syou Maki[1], Sachiko Omotani[1,2], Junji Mukai[1], Tomoya Tachi[3], Hitomi Teramachi[3], Michiaki Myotoku[1]

1 Faculty of Pharmacy, Osaka Ohtani University, Tondabayashi-shi, Osaka, Japan, 2 Sakai City Medical Center, Sakai-shi, Osaka, Japan, 3 Gifu Pharmaceutical University, Gifu-shi, Gifu, Japan

* hatuday@osaka-ohtani.ac.jp

**Data Availability Statement:** All relevant data are within the manuscript and its Supporting Information files.

## Abstract

### Background

To support effective antibiotic selection in empirical treatments, infection control interventions, and antimicrobial resistance containment strategies, many medical institutions collect antimicrobial susceptibility test data conducted at their facilities to prepare cumulative antibiograms.

### Aim

To evaluate how the setpoints of duplicate isolate removal period and data collection period affect the calculated susceptibility rates in antibiograms.

### Methods

The Sakai City Medical Center is a regional core hospital for tertiary emergency medical care with 480 beds for general clinical care. In this study, all the *Pseudomonas aeruginosa*, *Escherichia coli*, and *Klebsiella pneumoniae* isolates collected at the Sakai City Medical Center Clinical Laboratory between July 2013 and December 2018 were subjected to antimicrobial susceptibility tests and the resulting data was analyzed.

### Findings

The longer the duplicate isolate removal period, the fewer the isolates are available for every bacterial species. Differences in the length of the duplicate isolate removal period affected *P. aeruginosa* susceptibility rates to β-lactam antibiotics by up to 10.8%. The setpoint of the data collection period affected the antimicrobial susceptibility rates by up to 7.3%. We found that a significant change in susceptibility could be missed depending on the setting of the data collection period, in preparing antibiogram of β-lactam antibiotics for *P. aeruginosa*.

**Funding:** The authors received no specific funding for this work.

**Competing interests:** The authors have declared that no competing interests exist.

## Conclusions

When referring to antibiograms, medical professionals involved in infectious disease treatment should be aware that the parameter values, such as the duplicate isolate removal period and the data collection period, affect *P. aeruginosa* susceptibility rates especially to β-lactam antibiotics. And antibiogram should be updated within the shortest time period that is practically possible, taking into account restrictions such as numbers of specimen.

## Introduction

In light of the current global problem of increasing drug resistance, the importance of using antibiotics appropriately is now widely recognized [1]. It is also crucial to monitor trends in antimicrobial resistance at the facility or local level to support effective antibiotic selection in empirical treatments, infection control interventions, and antimicrobial resistance containment strategies. Therefore, many medical institutions collect antimicrobial susceptibility test data conducted at their facilities to calculate susceptibility rates and prepare cumulative antibiograms [2].

Guidelines have been issued by the Clinical and Laboratory Standards Institute (CLSI) for the preparation of such antibiograms [3]. These guidelines set out several recommendations including updating antibiogram data at least once per year, analyzing a minimum of 30 isolates per species, and using only the first isolate when there are repeat or multiple isolates from a patient. However, as compliance to these guidelines requires a significant amount of time and effort, many facilities use dedicated computer software to generate antibiograms [4].

While there have been several research studies on how the proposed conditions in the CLSI guidelines affect the calculated susceptibility rates in antibiograms [5, 6], no similar studies have been carried out in Japan.

Sakai City Medical Center is a regional core hospital with facilities for tertiary emergency medical care; there are 480 beds for general clinical care and 7 isolated units for patients with infectious diseases. On July 1 2015, the hospital relocated to a new location approximately 3.5 km away from its original location. The new location is part of a different primary medical care area.

In this study, we evaluated the impact on antimicrobial susceptibility rates when various setpoints for data collection period and duplicate isolate removal period were used during the preparation of cumulative antibiograms. Antimicrobial susceptibility test results, both prior to and after relocation of the Sakai City Medical Center to its new location were also compared.

## Materials and methods

### Ethics statement

This study was approved by the institutional review board of the Sakai City Medical Center (approval number: H30-119). All data was strictly protected after connectable anonymization, and there was no possibility of personal identification in the analyzing process.

### Target bacteria, survey period, antimicrobial agents, and determination of susceptibility rates

Gram-negative bacteria can cause many types of infection, including pneumonia, bloodstream infections, wound or surgical site infections, and meningitis; also, Gram-negative bacteria are

**Table 1. Target antimicrobial agents.**

| Penicillins [*] | | | Carbapenems [*] | |
|---|---|---|---|---|
| ampicillin | ABPC | | doripenem | DRPM |
| ampicillin/sulbactam | ABPC/SBT | | imipenem | IPM |
| piperacillin | PIPC | | meropenem | MEPM |
| piperacillin/tazobactam | PIPC/TAZ | | **Aminoglycosides** | |
| **Cephalosporins** [*] | | | amikacin | AMK |
| ceftazidime | CAZ | | gentamicin | GM |
| cefaclor | CCL | | tobramycin | TOB |
| cefazolin | CEZ | | **Tetracyclines** | |
| cefepime | CFPM | | minocycline | MINO |
| cefcapene pivoxil | CFPN | | **Fluoroquinolones** | |
| cefmetazole | CMZ | | ciprofloxacin | CPFX |
| cefoperazone/sulbactam | CPZ/SBT | | levofloxacin | LVFX |
| cefotiam | CTM | | **Polymyxins** | |
| ceftriaxone | CTRX | | colistin | CL |
| cefotaxime | CTX | | **Others** | |
| cefozopran | CZOP | | fosfomycin | FOM |
| flomoxef | FMOX | | sulfamethoxazole/trimethoprim | ST |
| **Monobactams** [*] | | | | |
| aztreonam | AZT | | | [*] β-lactam antibiotics |

becoming increasingly resistant to multiple drugs. Hence, among Gram-negative bacteria, *Pseudomonas aeruginosa*, *Klebsiella pneumoniae*, and *Escherichia coli*, which are frequently isolated, were chosen as the target species.

The survey period was five and a half years, spanning July 2013 to December 2018. All isolates collected at the Sakai City Medical Center Clinical Laboratory during this survey period were subjected to antimicrobial susceptibility tests and the resulting data was analyzed.

The target antimicrobial agents for drug susceptibility tests were penicillins, cephalosporins, carbapenems, monobactams, aminoglycosides, tetracyclines, fluoroquinolones, polymyxins, and others in routine use at the Sakai City Medical Center. Table 1 shows target antimicrobial agents by class and their respective abbreviations.

The MIC values from the antimicrobial susceptibility tests were classified as susceptible (S), intermediate (I), or resistant (R), according to the CLSI criteria (M100-S26) [7].

## Data

**Calculation of susceptibility rates.**   To calculate the susceptibility rates, we used software that we had specifically developed for creating antibiograms, Chans (<u>Ch</u>arts and <u>An</u>tibiogram Making <u>S</u>ystem). Chans is capable of creating antibiograms by setting various conditions such as the specified bacterial species, the antimicrobial agents, and the removal period of duplicate isolates, the data collection period, and the data collection interval [8].

In this study the S, I, and R results were uploaded into the Chans system, and the conditions such as the duplicate isolate removal period, the data collection period, and the data collection time were set. The susceptibility rates were then calculated using the following equation:

$$Susceptibility\ rate = \frac{no.of\ \mathbf{S}\ isolates}{no.of\ \mathbf{S}\ isolates + no.of\ \mathbf{I}\ isolates + no.of\ \mathbf{R}\ isolates}$$

**Data collection timing and period.** For the annual susceptibility rates, cumulative data was calculated in December, whilst this was done in June and December for the bi-annual analyses. The data collection period was defined as the last six months or one year, including the aggregate month.

**Handling of duplicate isolates.** As it is possible that isolates from the same patient among long-term in-patients may undergo multiple antimicrobial susceptibility tests during the study period, the CLSI guidelines recommend that only the first isolate is used for analysis per patient per reporting period for a particular antimicrobial agent to prevent results from being affected by duplicate isolates. While there are various ideas on how to deal with duplicate isolates [3, 5, 6], there has been no unified agreement on the removal period of duplicate isolates [5, 6].

In this study, the duplicate isolate removal periods were set at 365 days, 180 days, and 30 days and the data was collected, respectively.

**Number of patients targeted before and after hospital relocation.** With a duplicate isolate removal period of 30 days, the target patients for the last 6 months were analyzed every 6 months. After July 2015, when the hospital was relocated, patients on whom a susceptibility test had been performed at its previous site were identified by their patient IDs, and the proportion that they accounted for out of all the target patients was calculated.

**Statistical analysis.** Data were analyzed using R software. The chi-square test was used to assess for associations between susceptibility rate and the duplicate isolate removal periods. Differences in sensibility rates between the long and short data collection periods were also tested using the chi-square test.

## Results

### Duplicate isolate removal period

**Influence on number of excluded isolates.** For each bacterial species, aggregate data was compiled and analyzed at different duplicate isolate removal periods (30 days, 180 days, and 365 days) and was compared with when all the isolates were used. This data was compared annually. Table 2 shows the annual comparisons between the numbers of isolates, the number of excluded isolates, and their proportions at the specified duplicate isolate removal periods for each bacterial species.

For all bacterial species, the proportion of excluded isolates increased as the duplicate isolate removal period increased, with a concomitant reduction in the number of target isolates. The mean proportion and (standard deviation) of excluded isolates when the duplicate isolate removal periods were set at 365 days, 180 days, and 30 days were 42.0 (2.2), 40.7 (2.4), and 32.2 (2.9) for *P. aeruginosa*; 27.7 (1.6), 26.9 (1.5), and 22.1 (1.3) for *E. coli*; and 23.1 (2.8), 22.7 (2.7), and 18.7 (2.7) for *K. pneumoniae*, with no significant annual variations noted.

**Influence on susceptibility rates.** Tables 3–5 show the antimicrobial agent susceptibility rates for each bacterial species when the duplicate isolate removal periods were 365 days, 180 days, and 30 days, and when all the isolates were used. All susceptibility rates when the duplicate isolate removal periods were set were compared against when all strains were used. When there were significant differences, these were indicated by asterisks in the tables.

The *P. aeruginosa* susceptibility rates to β-lactam antibiotics (penicillins, cephalosporins, carbapenems, and monobactams) increased as the duplicate isolate removal period increased. When the duplicate isolate removal periods were 365 days and 180 days, the susceptibility rates for most β-lactam antibiotics were significantly higher compared with when all isolates were used.

By setting different duplicate isolate removal periods and comparing the data to when all the isolates were used, the biggest increase in drug susceptibility rates was observed for AZT in

**Table 2. Effect of duplicate removal period on the number of excluded isolates.**

| Year | Duplicate isolate removal period | Number of excluded isolates / Number of isolates | | | | | | | | | Percentage of excluded isolates (%) | | |
|---|---|---|---|---|---|---|---|---|---|---|---|---|---|
| | | *P. aeruginosa* | | | *E. coli* | | | *K. pneumoniae* | | | *P. aeruginosa* | *E. coli* | *K. pneumoniae* |
| 2014 | 365 days | 157 | / | 218 | 159 | / | 444 | 50 | / | 216 | 41.9 | 26.4 | 18.8 |
| | 180 days | 151 | / | 224 | 157 | / | 446 | 50 | / | 216 | 40.3 | 26.0 | 18.8 |
| | 30 days | 109 | / | 266 | 136 | / | 467 | 38 | / | 228 | 29.1 | 22.6 | 14.3 |
| | all isolates | 375 | | | 603 | | | 266 | | | - | - | - |
| 2015 | 365 days | 180 | / | 241 | 251 | / | 641 | 66 | / | 219 | 42.8 | 28.1 | 23.2 |
| | 180 days | 173 | / | 248 | 241 | / | 651 | 65 | / | 220 | 41.1 | 27.0 | 22.8 |
| | 30 days | 139 | / | 282 | 194 | / | 698 | 53 | / | 232 | 33.0 | 21.7 | 18.6 |
| | all isolates | 421 | | | 892 | | | 285 | | | - | - | - |
| 2016 | 365 days | 177 | / | 264 | 319 | / | 805 | 79 | / | 291 | 40.1 | 28.4 | 21.4 |
| | 180 days | 171 | / | 270 | 307 | / | 817 | 76 | / | 294 | 38.8 | 27.3 | 20.5 |
| | 30 days | 132 | / | 309 | 241 | / | 883 | 65 | / | 305 | 29.9 | 21.4 | 17.6 |
| | all isolates | 441 | | | 1124 | | | 370 | | | - | - | - |
| 2017 | 365 days | 200 | / | 306 | 323 | / | 751 | 114 | / | 328 | 39.5 | 30.1 | 25.8 |
| | 180 days | 194 | / | 312 | 314 | / | 760 | 112 | / | 330 | 38.3 | 29.2 | 25.3 |
| | 30 days | 159 | / | 347 | 261 | / | 813 | 97 | / | 345 | 31.4 | 24.3 | 21.9 |
| | all isolates | 506 | | | 1074 | | | 442 | | | - | - | - |
| 2018 | 365 days | 233 | / | 278 | 269 | / | 781 | 121 | / | 341 | 45.6 | 25.6 | 26.2 |
| | 180 days | 231 | / | 280 | 261 | / | 789 | 120 | / | 342 | 45.2 | 24.9 | 26.0 |
| | 30 days | 191 | / | 320 | 214 | / | 836 | 98 | / | 364 | 37.4 | 20.4 | 21.2 |
| | all isolates | 511 | | | 1050 | | | 462 | | | - | - | - |

2014, with values of 74.1% (all isolates) to 84.9% (at 365 days). Susceptibility rates with the smallest increases were observed in 2016 for DRPM, with values of 93.4%–94.8%.

The *P. aeruginosa* susceptibility rates to non-β-lactam antibiotics did not show a regular increase or decrease with changes in the duplicate isolate removal periods.

The *E. coli* susceptibility rates to all antimicrobial agents did not show a regular increase or decrease with the change in the duplicate isolate removal periods. In addition, there was no significant change in the susceptibility rate due to the duplicate isolate removal periods. However, with CCL alone, the susceptibility rates increased as the duplicate isolate removal periods increased, albeit with only a small change of up to 3%.

The *K. pneumoniae* susceptibility rates did not show a regular trend for all antimicrobial agents. There also was no significant change in the susceptibility rate due to the duplicate isolate removal periods. However, in the case of CEZ alone, the susceptibility rates increased as the duplicate isolate removal periods increased, although the change range was as small as 1.5%.

Please refer to the supporting information for detailed data on the difference in susceptibility when all isolates were used and when the duplicate isolate removal period was 365 days, 180 days, and 30 days (S1 Table).

In order to compare the variation in susceptibility rate due to the duplicate isolate removal periods among antimicrobial agents, the susceptibility rate change resulting from the duplicate isolate removal periods was aggregated every year and the coefficient of variation was calculated. This is shown in Figs 1–3 for each bacterial species.

The coefficients of variation of susceptibility rates of each antimicrobial agent by the duplicate isolate removal periods in *P. aeruginosa* were one order of magnitude greater than in *E.*

**Table 3. Effect of duplicate isolate removal period on susceptible rates of *P. aeruginosa*.**

| Year | Dupulicate isolate removal period | Susceptible rate (%) | | | | | | | | | | | | | | | | |
|---|---|---|---|---|---|---|---|---|---|---|---|---|---|---|---|---|---|---|
| | | PIPC | PIPC/TAZ | CAZ | CFPM | CPZ/SBT | CZOP | DRPM | IPM | MEPM | AZT | AMK | GM | TOB | CPFX | LVFX | CL | FOM |
| 2014 | 365 days | 88.5** | 93.1** | 91.3** | 90.4** | 89.9* | 94.0 | 95.4** | 88.1* | 92.2* | 84.9* | 94.0 | 79.4 | 95.4 | 92.2 | 91.3** | 98.6 | 9.2 |
| | 180 days | 87.9** | 92.9** | 90.6** | 89.7** | 89.3* | 94.6 | 95.1** | 87.9* | 91.0* | 84.4* | 94.2 | 78.6 | 95.5 | 91.0 | 90.6** | 98.7 | 8.9 |
| | 30 days | 85.7 | 90.2 | 87.0 | 86.8 | 86.5 | 91.7 | 92.9 | 86.1** | 89.8 | 81.2** | 94.4 | 77.1 | 95.1 | 91.4 | 88.3 | 98.5 | 9.0 |
| | all isolates | 80.8 | 86.4 | 84.5 | 82.4 | 80.5 | 90.1 | 89.1 | 79.2 | 84.3 | 74.1 | 94.9 | 76.8 | 96.0 | 88.8 | 84.3 | 98.1 | 9.6 |
| 2015 | 365 days | 95.4* | 97.5* | 97.9* | 95.0** | 95.0** | 98.3* | 97.9* | 88.4 | 95.0** | 90.5 | 97.5 | 86.7 | 99.2 | 94.6 | 93.8 | 97.9 | 8.7 |
| | 180 days | 95.6* | 97.6* | 97.0* | 95.2** | 95.2** | 98.4* | 97.0* | 88.3 | 95.2** | 90.7 | 97.6 | 87.1 | 99.2 | 94.8 | 93.5 | 97.0 | 8.5 |
| | 30 days | 92.9 | 95.0 | 94.7** | 92.2 | 91.5 | 95.7 | 95.4 | 85.8 | 92.2 | 87.6 | 97.2 | 86.9 | 98.6 | 92.9 | 91.5 | 98.2 | 8.2 |
| | all isolates | 88.6 | 90.0 | 90.0 | 89.1 | 89.1 | 92.9 | 92.6 | 83.6 | 89.5 | 85.5 | 97.1 | 85.5 | 98.3 | 90.0 | 90.0 | 98.3 | 7.1 |
| 2016 | 365 days | 97.7* | 99.2* | 98.5* | 96.0* | 95.5* | 98.9* | 97.3** | 90.2** | 93.2** | 90.9** | 96.6 | 90.2 | 97.3 | 93.9 | 93.9 | 98.5 | 7.2 |
| | 180 days | 96.7* | 98.9* | 98.5* | 96.3* | 95.2* | 98.9* | 97.0 | 90.0** | 92.0** | 90.4 | 96.7 | 89.6 | 97.4 | 93.3 | 93.3 | 98.5 | 7.0 |
| | 30 days | 92.6 | 95.5 | 94.5 | 92.2 | 91.3 | 96.4 | 94.8 | 88.0 | 90.9 | 86.7 | 96.1 | 87.7 | 97.1 | 91.3 | 90.9 | 98.4 | 6.5 |
| | all isolates | 89.3 | 92.1 | 91.6 | 90.5 | 88.4 | 94.3 | 93.4 | 84.1 | 87.0 | 85.3 | 96.8 | 89.3 | 97.7 | 91.4 | 90.9 | 98.9 | 5.7 |
| 2017 | 365 days | 93.8** | 95.1** | 94.8** | 95.1** | 93.1** | 97.4** | 96.7 | 93.1 | 94.8 | 89.2** | 98.4 | 92.5 | 99.0 | 94.1 | 95.4 | 98.7 | 5.9 |
| | 180 days | 93.6** | 95.2** | 94.9** | 94.9** | 92.9** | 97.4** | 96.5 | 92.9 | 94.6 | 89.1** | 98.4 | 92.3 | 99.0 | 93.9 | 95.2 | 98.7 | 5.8 |
| | 30 days | 92.5** | 94.5** | 94.2 | 93.9 | 91.6 | 96.5 | 95.7 | 92.2 | 93.7 | 87.9 | 98.6 | 92.2 | 99.1 | 93.9 | 94.5 | 98.6 | 6.3 |
| | all isolates | 88.1 | 90.1 | 90.7 | 90.3 | 87.7 | 94.1 | 94.3 | 90.7 | 92.3 | 83.0 | 98.6 | 91.9 | 99.2 | 95.5 | 95.7 | 98.2 | 7.7 |
| 2018 | 365 days | 95.3* | 95.7* | 96.4* | 94.2** | 94.0** | 97.5** | 98.2* | 91.0* | 97.1* | 88.5 | 98.6 | 85.6 | 99.3 | 94.2 | 93.2 | 98.9 | 6.8 |
| | 180 days | 95.4* | 95.7* | 96.1* | 94.3** | 95.0** | 97.5** | 98.2* | 90.7* | 97.1* | 88.2 | 98.6 | 85.7 | 99.3 | 93.9 | 92.9 | 98.9 | 6.8 |
| | 30 days | 93.4 | 93.8 | 94.7** | 92.8 | 93.1 | 96.3 | 97.2** | 89.1** | 95.0* | 85.9 | 98.8 | 85.9 | 99.4 | 93.8 | 92.5 | 98.8 | 6.3 |
| | all isolates | 89.6 | 90.0 | 90.4 | 89.2 | 90.0 | 93.3 | 93.2 | 83.8 | 89.4 | 83.4 | 98.6 | 85.7 | 99.6 | 93.5 | 92.0 | 98.6 | 9.4 |

365 days, 180 days, 30 days vs all isolates:

* $P < 0.01$

** $P < 0.05$

*coli* and *K. pneumoniae*. In *P. aeruginosa*, the coefficients of variation of susceptibility rates of β-lactam antibiotics were around 0.04, which were larger than those of other antimicrobials, and of aminoglycoside agents and colistin were much smaller (approximately 0.003). The coefficients of variation for fluoroquinolone agents were approximately 0.02, which were about halfway between the two, and the FOM showed larger coefficients of variation than the others.

In *E. coli*, the coefficients of variation in the susceptibility rates of each antimicrobial agent depending on the duplicate isolate removal periods were relatively high for β-lactam antibiotics, but most were less than 0.02, which was comparable to the coefficients of variation shown by aminoglycoside agents in *P. aeruginosa*. Among β-lactam antibiotics, PIPC/TAZ, CMZ, FMOX, and MEPM had low coefficients of variation of less than 0.005. The aminoglycoside agents AMK and GM also showed low values, whereas the fluoroquinolone LVFX showed relatively high values.

In *K. pneumoniae*, the coefficients of variation of the susceptibility rates of each antimicrobial by the duplicate isolate removal periods were generally even lower than those of *E. coli*, and many were less than 0.01. CMZ, IPM, MEPM, and AMK showed particularly low values, but these were the antimicrobials that always showed 100% or near 100% susceptibility.

**Table 4. Effect of duplicate isolate removal period on susceptible rates of *E. coli*.**

| Year | Dupulicate isolate removal period | Susceptible rate (%) | | | | | | | | | | | | | | | | | | | | | | | | |
|---|---|---|---|---|---|---|---|---|---|---|---|---|---|---|---|---|---|---|---|---|---|---|---|---|---|---|
| | | ABPC | ABPC/SBT | PIPC | PIPC/TAZ | CAZ | CCL | CEZ | CFPM | CFPN | CMZ | CPZ/SBT | CTM | CTRX | CTX | FMOX | IPM | MEPM | AZT | AMK | GM | MINO | LVFX | FOM | ST |
| 2014 | 365 days | 59.2 | 64.2 | 63.3 | 98.2 | 86.3 | 79.5 | 78.6 | 87.2 | 86.7 | 99.1 | 93.0 | 85.1 | 85.4 | 85.1 | 98.6 | 100.0 | 100.0 | 86.3 | 99.5 | 91.2 | 91.7 | 72.0 | 86.7 | 82.4 |
| | 180 days | 59.2 | 64.1 | 63.2 | 98.2 | 86.1 | 79.4 | 78.5 | 86.0 | 86.6 | 99.1 | 93.0 | 84.0 | 85.2 | 84.0 | 98.7 | 100.0 | 100.0 | 86.1 | 99.6 | 91.3 | 91.5 | 72.6 | 86.8 | 82.3 |
| | 30 days | 57.8 | 62.7 | 61.9 | 98.1 | 85.0 | 78.2 | 77.1 | 86.5 | 85.8 | 98.9 | 92.5 | 83.9 | 84.2 | 83.9 | 98.1 | 100.0 | 100.0 | 85.0 | 99.6 | 91.2 | 91.4 | 71.7 | 86.7 | 82.0 |
| | all isolates | 58.0 | 62.7 | 61.7 | 97.5 | 85.4 | 77.9 | 77.4 | 86.6 | 85.0 | 99.2 | 91.2 | 84.2 | 84.6 | 84.4 | 98.3 | 100.0 | 100.0 | 85.4 | 99.7 | 91.0 | 91.9 | 72.1 | 87.4 | 81.6 |
| 2015 | 365 days | 54.1 | 64.4 | 60.2 | 98.4 | 80.5 | 73.9 | 73.6 | 81.1 | 80.3 | 99.1 | 93.8 | 79.7 | 79.9 | 79.9 | 98.6 | 100.0 | 100.0 | 79.7 | 99.7 | 89.4 | 90.5 | 68.5 | 87.4 | 79.4 |
| | 180 days | 53.9 | 64.4 | 59.9 | 98.5 | 80.2 | 73.7 | 73.4 | 80.8 | 80.1 | 99.1 | 93.7 | 79.4 | 79.6 | 79.6 | 98.6 | 100.0 | 100.0 | 79.4 | 99.7 | 89.1 | 90.5 | 68.0 | 87.4 | 79.1 |
| | 30 days | 52.4 | 63.2 | 58.6 | 98.4 | 78.9 | 72.3 | 72.5 | 79.5 | 78.6 | 99.1 | 92.6 | 78.2 | 78.4 | 78.4 | 98.6 | 100.0 | 100.0 | 78.2 | 99.7 | 88.0 | 89.3 | 66.0 | 87.5 | 77.8 |
| | all isolates | 52.7 | 63.0 | 58.4 | 98.3 | 78.6 | 71.0 | 72.4 | 79.4 | 78.2 | 98.0 | 92.2 | 77.8 | 78.1 | 78.1 | 98.3 | 100.0 | 100.0 | 78.0 | 99.7 | 89.0 | 89.2 | 64.6 | 88.6 | 77.8 |
| 2016 | 365 days | 51.9 | 60.9 | 58.4 | 97.3 | 78.4 | 69.9 | 70.2 | 79.4 | 77.8 | 98.8 | 91.3 | 77.5 | 77.9 | 77.6 | 98.3 | 99.9 | 99.8 | 78.6 | 99.6 | 90.5 | 89.2 | 66.0 | 91.1 | 79.4 |
| | 180 days | 51.7 | 60.0 | 58.1 | 97.3 | 78.0 | 69.5 | 69.9 | 79.1 | 77.4 | 98.8 | 91.2 | 77.1 | 77.5 | 77.2 | 98.3 | 99.9 | 99.8 | 78.1 | 99.6 | 90.5 | 89.1 | 65.8 | 91.2 | 79.4 |
| | 30 days | 50.4 | 60.6 | 56.8 | 97.3 | 76.8 | 68.3 | 68.6 | 77.0 | 76.0 | 98.8 | 90.8 | 75.9 | 76.3 | 75.0 | 98.2 | 99.9 | 99.8 | 76.9 | 99.7 | 90.6 | 88.4 | 64.6 | 90.8 | 78.4 |
| | all isolates | 50.7 | 61.2 | 56.8 | 96.4 | 75.6 | 67.5 | 67.9 | 76.6 | 74.4 | 98.7 | 90.5 | 74.7 | 75.1 | 74.9 | 98.4 | 99.9 | 99.8 | 75.7 | 99.7 | 90.9 | 88.6 | 63.9 | 91.2 | 78.8 |
| 2017 | 365 days | 55.3 | 64.3 | 60.7 | 98.9 | 83.1 | 76.8 | 77.5 | 84.3 | 83.3 | 99.2 | 94.7 | 82.8 | 83.1 | 82.7 | 98.9 | 100.0 | 100.0 | 83.4 | 99.9 | 92.4 | 93.5 | 69.9 | 90.9 | 83.2 |
| | 180 days | 54.9 | 63.9 | 60.3 | 98.9 | 82.9 | 76.7 | 77.2 | 84.1 | 83.0 | 99.2 | 94.6 | 82.6 | 82.9 | 82.5 | 98.9 | 100.0 | 100.0 | 83.2 | 99.9 | 92.2 | 93.4 | 69.5 | 90.8 | 82.9 |
| | 30 days | 53.6 | 63.1 | 58.8 | 98.9 | 81.1 | 75.2 | 75.8 | 82.3 | 81.1 | 99.3 | 93.7 | 80.8 | 81.1 | 80.7 | 99.0 | 100.0 | 100.0 | 81.4 | 99.9 | 92.6 | 93.1 | 67.5 | 90.5 | 81.9 |
| | all isolates | 52.0 | 61.5 | 56.9 | 98.7 | 79.5 | 73.8 | 74.6 | 80.7 | 79.4 | 99.3 | 93.5 | 79.3 | 79.6 | 79.2 | 99.2 | 100.0 | 100.0 | 79.9 | 99.9 | 92.2 | 92.7 | 66.4 | 91.5 | 81.2 |
| 2018 | 365 days | 53.8 | 63.4 | 58.3 | 98.2 | 80.9 | 73.5 | 73.5 | 81.9 | 81.2 | 98.7 | 93.0 | 80.0 | 80.3 | 80.0 | 98.2 | 99.9 | 99.9 | 80.8 | 99.7 | 90.3 | 93.5 | 66.2 | 92.1 | 78.9 |
| | 180 days | 53.6 | 63.1 | 58.2 | 98.1 | 80.0 | 73.4 | 73.4 | 82.0 | 81.2 | 98.7 | 93.9 | 80.1 | 80.4 | 80.1 | 98.2 | 99.9 | 99.9 | 80.9 | 99.7 | 90.4 | 93.5 | 66.2 | 92.1 | 78.8 |
| | 30 days | 52.9 | 62.6 | 57.8 | 97.0 | 80.9 | 72.7 | 72.7 | 81.8 | 81.0 | 98.6 | 93.7 | 79.8 | 80.1 | 79.8 | 98.1 | 99.9 | 99.9 | 80.6 | 99.8 | 90.7 | 93.4 | 65.7 | 91.9 | 78.6 |
| | all isolates | 51.5 | 61.7 | 56.2 | 98.2 | 79.0 | 70.8 | 70.8 | 80.2 | 79.1 | 98.6 | 93.7 | 78.0 | 78.4 | 78.1 | 97.9 | 99.9 | 99.8 | 78.8 | 99.8 | 90.8 | 93.0 | 65.2 | 92.0 | 77.7 |

365 days, 180 days, 30 days vs all isolates: * $P<0.01$, ** $P<0.05$

**Table 5. Effect of duplicate isolate removal period on susceptible rates of *K. pneumoniae*.**

| Year | Dupulicate isolate removal period | ABPC | ABPC/SBT | PIPC | PIPC/TAZ | CAZ | CCL | CEZ | CFPM | CFPN | CMZ | CPZ/SBT | CTM | CTRX | CTX | FMOX | IPM | MEPM | AZT | AMK | GM | MINO | LVFX | FOM | ST |
|---|---|---|---|---|---|---|---|---|---|---|---|---|---|---|---|---|---|---|---|---|---|---|---|---|---|
| | | | | | | | | | | | | | | | | | | | | Susceptible rate (%) | | | | | |
| 2014 | 365 days | 0.0 | 84.7 | 62.5 | 98.6 | 95.8 | 94.4 | 93.1 | 95.8 | 95.4 | 99.1 | 97.7 | 94.0 | 95.4 | 95.4 | 99.5 | 100.0 | 100.0 | 95.8 | 100.0 | 98.1 | 85.2 | 98.1 | 35.2 | 90.3 |
| | 180 days | 0.0 | 84.7 | 62.5 | 98.6 | 95.8 | 94.4 | 93.1 | 95.8 | 95.4 | 99.1 | 97.7 | 94.0 | 95.4 | 95.4 | 99.5 | 100.0 | 100.0 | 95.8 | 100.0 | 98.1 | 85.2 | 98.1 | 35.2 | 90.3 |
| | 30 days | 0.0 | 83.8 | 61.8 | 98.2 | 94.7 | 93.4 | 92.1 | 94.7 | 94.3 | 99.1 | 96.9 | 93.0 | 94.3 | 94.3 | 99.6 | 100.0 | 100.0 | 94.7 | 100.0 | 97.4 | 84.6 | 98.2 | 34.6 | 89.5 |
| | all isolates | 0.0 | 83.8 | 63.2 | 98.1 | 94.4 | 93.2 | 92.1 | 94.4 | 93.9 | 99.2 | 96.6 | 92.9 | 94.0 | 94.0 | 99.6 | 100.0 | 100.0 | 94.4 | 100.0 | 97.0 | 85.3 | 98.1 | 34.2 | 89.1 |
| 2015 | 365 days | 0.0 | 91.8 | 71.2 | 98.2 | 97.7 | 95.9 | 95.9 | 98.2 | 98.1 | 100.0 | 98.6 | 97.7 | 97.7 | 97.7 | 99.5 | 100.0 | 100.0 | 97.7 | 100.0 | 99.1 | 92.2 | 99.5 | 39.7 | 93.6 |
| | 180 days | 0.0 | 91.8 | 71.4 | 98.2 | 97.7 | 95.9 | 95.9 | 98.2 | 98.1 | 100.0 | 98.6 | 97.7 | 97.7 | 97.7 | 99.5 | 100.0 | 100.0 | 97.7 | 100.0 | 99.1 | 92.3 | 99.5 | 40.0 | 93.6 |
| | 30 days | 0.0 | 90.9 | 71.1 | 98.3 | 97.4 | 95.7 | 95.7 | 97.8 | 97.8 | 100.0 | 98.7 | 97.4 | 97.4 | 97.4 | 99.6 | 100.0 | 100.0 | 97.4 | 100.0 | 98.7 | 91.8 | 99.6 | 40.9 | 93.5 |
| | all isolates | 0.0 | 90.5 | 70.2 | 97.9 | 97.2 | 95.8 | 95.4 | 98.2 | 98.2 | 100.0 | 98.2 | 97.9 | 97.9 | 97.9 | 99.6 | 100.0 | 100.0 | 97.5 | 100.0 | 98.9 | 90.9 | 98.9 | 41.4 | 93.7 |
| 2016 | 365 days | 0.0 | 90.4 | 70.1 | 99.0 | 96.9 | 95.9 | 95.5 | 96.9 | 96.7 | 99.3 | 98.6 | 95.9 | 96.9 | 96.6 | 99.3 | 99.7 | 99.7 | 96.9 | 100.0 | 97.9 | 90.4 | 98.3 | 31.3 | 93.8 |
| | 180 days | 0.0 | 90.1 | 70.1 | 99.0 | 96.9 | 95.9 | 95.6 | 96.9 | 96.8 | 99.3 | 98.6 | 95.9 | 96.9 | 96.6 | 99.3 | 99.7 | 99.7 | 96.9 | 100.0 | 97.6 | 90.5 | 98.3 | 31.3 | 93.5 |
| | 30 days | 0.0 | 90.5 | 70.8 | 99.0 | 97.0 | 96.1 | 95.7 | 97.0 | 96.9 | 99.3 | 98.7 | 96.1 | 97.0 | 96.7 | 99.3 | 99.7 | 99.7 | 97.0 | 100.0 | 97.7 | 90.5 | 98.4 | 31.8 | 93.4 |
| | all isolates | 0.0 | 90.0 | 70.0 | 98.4 | 96.5 | 95.7 | 95.4 | 96.5 | 97.1 | 98.4 | 97.8 | 95.7 | 96.5 | 96.2 | 98.4 | 99.2 | 99.2 | 96.5 | 100.0 | 97.8 | 89.7 | 98.4 | 33.2 | 93.0 |
| 2017 | 365 days | 0.0 | 85.1 | 71.3 | 98.8 | 96.6 | 95.1 | 95.1 | 96.6 | 96.5 | 99.7 | 97.9 | 95.7 | 96.3 | 95.7 | 99.7 | 100.0 | 100.0 | 96.3 | 100.0 | 96.3 | 88.1 | 98.5 | 30.2 | 93.3 |
| | 180 days | 0.0 | 84.8 | 71.2 | 98.8 | 96.7 | 95.2 | 95.2 | 96.7 | 96.5 | 99.7 | 97.9 | 95.8 | 96.4 | 95.8 | 99.7 | 100.0 | 100.0 | 96.4 | 100.0 | 96.4 | 88.2 | 98.5 | 30.3 | 93.3 |
| | 30 days | 0.0 | 84.1 | 70.4 | 98.8 | 95.9 | 94.8 | 94.5 | 96.2 | 96.0 | 99.4 | 98.0 | 95.1 | 95.9 | 95.4 | 99.7 | 100.0 | 100.0 | 95.9 | 100.0 | 96.5 | 87.5 | 98.6 | 30.1 | 93.0 |
| | all isolates | 0.0 | 82.4 | 70.4 | 98.9 | 95.7 | 94.1 | 93.7 | 95.7 | 95.7 | 99.1 | 98.0 | 94.3 | 95.5 | 94.8 | 99.3 | 100.0 | 100.0 | 95.5 | 100.0 | 96.4 | 88.5 | 98.4 | 30.3 | 92.8 |
| 2018 | 365 days | 0.0 | 84.5 | 59.8 | 98.2 | 96.2 | 95.0 | 94.7 | 96.8 | 96.6 | 99.1 | 99.4 | 95.9 | 96.2 | 95.9 | 99.1 | 99.7 | 100.0 | 96.2 | 100.0 | 97.9 | 89.7 | 97.4 | 24.3 | 91.5 |
| | 180 days | 0.0 | 84.2 | 59.6 | 98.2 | 95.9 | 94.7 | 94.4 | 96.5 | 96.3 | 99.1 | 99.4 | 95.6 | 95.9 | 95.6 | 99.1 | 99.7 | 100.0 | 95.9 | 100.0 | 97.7 | 89.5 | 97.4 | 24.6 | 91.2 |
| | 30 days | 0.0 | 83.2 | 59.1 | 97.8 | 95.3 | 94.2 | 93.7 | 96.2 | 95.9 | 98.9 | 98.9 | 95.1 | 95.3 | 95.1 | 98.9 | 99.7 | 100.0 | 95.3 | 100.0 | 97.8 | 89.0 | 97.3 | 24.2 | 90.4 |
| | all isolates | 0.0 | 82.7 | 57.8 | 98.3 | 95.5 | 94.2 | 93.5 | 96.3 | 95.9 | 98.3 | 99.1 | 94.8 | 95.7 | 95.0 | 98.5 | 99.6 | 100.0 | 95.7 | 100.0 | 98.1 | 87.9 | 97.8 | 25.1 | 90.9 |

365 days, 180 days, 30 days vs all isolates: * *P*<0.01, ** *P*<0.05

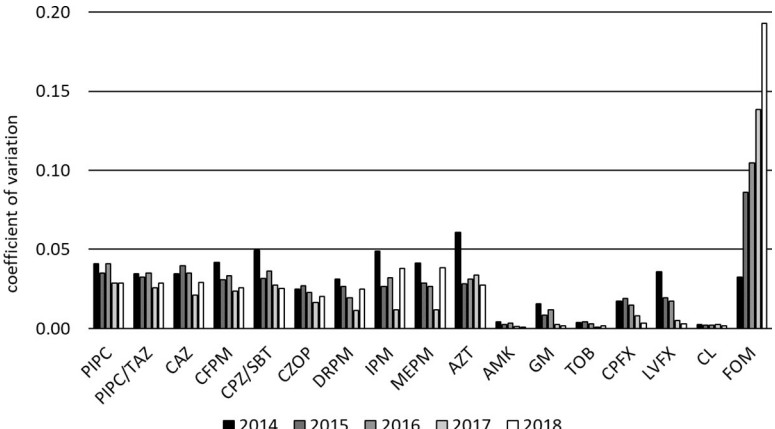

**Fig 1. Coefficients of variation of *P. aeruginosa* susceptibility rates to each antibacterial drug due to the duplicate isolate removal periods for each year.**

### Data collection period

Antimicrobial susceptibility rates were stratified every 6 months according to duplicate isolate removal periods and data collection periods of 6 months or 1 year over the course of the study. The *P. aeruginosa* susceptibility rates to β-lactam antibiotics, as well as the average values for the five-year study period are shown in Table 6.

The susceptibility of *P. aeruginosa* to β-lactam antibiotics increases or decreases depending on the data collection period, and no regularity was found. In terms of variation in the data collection periods, the highest observed difference in susceptibility rates was for PIPC in June 2014, and this was consistent at all duplicate isolate removal periods. At the 180-day duplicate isolate removal period, the rates for PIPC were 84.7% for the 6-month data collection period and 92.0% for the 12-month data collection period, indicating a difference of 7.3%. Similarly, the PIPC rates at the 30-day duplicate isolate removal period were 82.8% (6-month periods) and 89.9% (12-month periods), indicating a difference of 7.1%. The PIPC rates when all isolates were used were 78.4% (6-month periods) and 87.2% (12-month periods), with a difference of 8.8%.

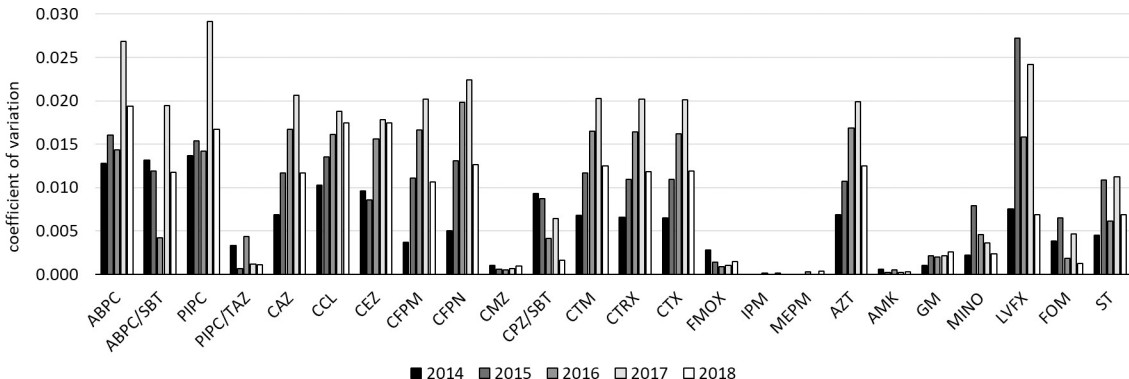

**Fig 2. Coefficients of variation of *E. coli* susceptibility rates to each antibacterial drug due to the duplicate isolate removal periods for each year.**

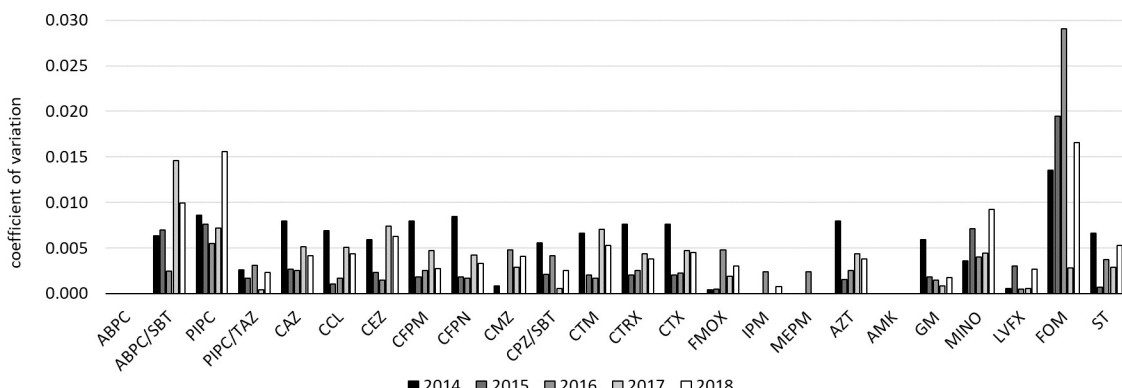

**Fig 3. Coefficients of variation of *K. pneumoniae* susceptibility rates to each antibacterial drug due to the duplicate isolate removal periods for each year.**

Significant differences in susceptibility rates due to data collection periods were observed for PIPC, PIPC/TAZ, and CAZ, all of which were collected in December 2015 with a 30-day duplicate isolate removal period, with susceptibility rates for the 6-month collection period being higher than those for the 1-year collection period ($P<0.05$). The susceptibility rates of PIPCs collected in January 2014 using all isolates also showed significant differences in susceptibility rates between different data collection periods ($P<0.05$).

In January 2014, the *P. aeruginosa* susceptibility rates were higher for all β-lactam antibiotics when the data collection period was set to 1 year compared with when it was set to 6 months. In addition, in December 2015, the *P. aeruginosa* susceptibility rates were lower for all β-lactam antibiotics when the data collection period was 1 year compared with when the data collection period was 6 months.

Fig 4 shows susceptibility rate variations for *P. aeruginosa* to PIPC, PIPC/TAZ, and CZOP when the data collection intervals were set at 6 months and 12 months for the duplicate isolate removal periods of 180 days and 30 days, or when all the isolates were used.

When the data collection period was 6 months, greater than 10% increases were noted in susceptibility rates for PIPC, PIPC/TAZ, and CZOP in the second half of 2015, followed by a more than 5% decline, with no further variations. Meanwhile, when the data collection period was 12 months, an approximately 5% increase in the susceptibility rate in the second half of 2015, nor any subsequent declines, were observed (Fig 4).

Please refer to the supporting information for detailed data about changes in the *P. aeruginosa* susceptibility rates to β-lactam antibiotics according to the data collection periods (S1 Fig).

The *P. aeruginosa* susceptibility rates to non-β-lactam antibiotics did not show regular changes due to the data collection period. The antimicrobial agent with the largest change in susceptibility rate due to differences in the data collection period was GM, which was calculated in June 2014, with the duplicate isolate removal period set to 30 days, with an 8.2% difference. In June 2014, susceptibility rates for all antimicrobial agents were higher when the data collection period was 1 year compared with when the data collection period was 6 months.

The *E. coli* susceptibility rates to β-lactam antibiotics did not show regular changes due to the data collection period. The antimicrobial agent with the largest change in susceptibility rate due to differences in the data collection period was CFPN, which was calculated in June 2014 using all isolates (a 7.1% difference).

**Table 6. Effect of the data collection period on *P. aeruginosa* susceptibility rates.**

| Antimicrobials | Interval considered as duplicate isolates | Collection period | Susceptible rate (%) | | | | | | | | | |
|---|---|---|---|---|---|---|---|---|---|---|---|---|
| | | | 2014 | | 2015 | | 2016 | | 2017 | | 2018 | |
| | | | Jun | Dec | Jun | Dec | Jun | Dec | Jun | Dec | Jun | Dec |
| PIPC | 180 days | 1 year | 92.0 | 87.9 | 90.5 | 95.6 | 97.6 | 96.7 | 94.0 | 93.6 | 95.1 | 95.4 |
| | | 6 months | 84.7 | 90.2 | 90.7 | 98.7 | 96.2 | 95.4 | 92.7 | 93.5 | 96.1 | 94.9 |
| | 30 days | 1 year | 89.9 | 85.7 | 87.3 | 92.9 | 94.8 | 92.6 | 92.5 | 92.5 | 93.5 | 93.4 |
| | | 6 months | 82.8 | 87.5 | 85.6 | 97.7* | 90.6 | 92.8 | 92.2 | 92.9 | 93.2 | 93.8 |
| | all isolates | 1 year | 87.2 | 80.8 | 82.4 | 88.6 | 92.1 | 89.3 | 88.8 | 88.1 | 87.8 | 89.6 |
| | | 6 months | 78.4* | 82.8 | 81.8 | 93.2 | 90.3 | 88.9 | 88.8 | 87.6 | 88.0 | 91.1 |
| PIPC/TAZ | 180 days | 1 year | 95.0 | 92.9 | 94.1 | 97.6 | 99.2 | 98.9 | 96.2 | 95.2 | 95.8 | 95.7 |
| | | 6 months | 89.8 | 94.7 | 93.8 | 100.0 | 98.1 | 97.7 | 94.7 | 95.2 | 95.3 | 95.5 |
| | 30 days | 1 year | 92.3 | 90.2 | 90.8 | 95.0 | 97.2 | 95.5 | 94.7 | 94.5 | 94.7 | 93.8 |
| | | 6 months | 87.1 | 92.1 | 88.3 | 99.4* | 94.0 | 95.4 | 94.0 | 95.1 | 93.2 | 94.3 |
| | all isolates | 1 year | 90.3 | 86.4 | 86.9 | 91.0 | 94.1 | 92.1 | 91.3 | 90.1 | 89.0 | 90.0 |
| | | 6 months | 84.2 | 88.2 | 85.3 | 94.8 | 92.9 | 91.6 | 90.8 | 89.5 | 88.4 | 91.4 |
| CAZ | 180 days | 1 year | 93.0 | 90.6 | 93.7 | 98.0 | 99.2 | 98.5 | 95.6 | 94.9 | 96.1 | 96.1 |
| | | 6 months | 86.7 | 93.2 | 94.8 | 100.0 | 98.1 | 97.1 | 94.0 | 95.2 | 96.1 | 96.2 |
| | 30 days | 1 year | 89.9 | 88.0 | 89.6 | 94.7 | 96.5 | 94.5 | 93.9 | 94.2 | 95.4 | 94.7 |
| | | 6 months | 84.5 | 90.1 | 88.3 | 98.8* | 93.2 | 94.3 | 93.4 | 95.1 | 94.5 | 94.9 |
| | all isolates | 1 year | 86.2 | 84.5 | 85.3 | 90.0 | 93.6 | 91.6 | 91.5 | 90.7 | 89.4 | 90.4 |
| | | 6 months | 82.5 | 86.3 | 84.1 | 94.0 | 92.9 | 90.9 | 92.1 | 89.5 | 89.3 | 91.4 |
| CFPM | 180 days | 1 year | 92.0 | 89.7 | 92.3 | 95.2 | 96.0 | 96.3 | 95.0 | 94.9 | 96.5 | 94.3 |
| | | 6 months | 85.7 | 92.4 | 91.8 | 97.4 | 94.3 | 96.5 | 93.4 | 95.8 | 94.5 | 94.3 |
| | 30 days | 1 year | 87.1 | 86.8 | 89.2 | 92.2 | 92.7 | 92.2 | 92.8 | 93.9 | 95.0 | 92.8 |
| | | 6 months | 81.9 | 90.1 | 87.4 | 95.3 | 88.9 | 93.3 | 92.2 | 95.6 | 92.5 | 93.2 |
| | all isolates | 1 year | 81.8 | 82.4 | 86.4 | 89.1 | 90.1 | 90.5 | 91.1 | 90.3 | 89.2 | 89.2 |
| | | 6 months | 77.8 | 86.3 | 86.5 | 90.8 | 89.0 | 91.3 | 90.8 | 89.8 | 88.4 | 90.0 |
| CPZ/SBT | 180 days | 1 year | 91.0 | 89.3 | 91.4 | 95.2 | 96.4 | 95.2 | 93.4 | 92.9 | 94.0 | 95.0 |
| | | 6 months | 86.7 | 90.9 | 90.7 | 98.0 | 94.3 | 94.2 | 92.7 | 92.9 | 94.5 | 95.5 |
| | 30 days | 1 year | 88.3 | 86.5 | 87.7 | 91.5 | 93.7 | 91.3 | 91.1 | 91.6 | 93.2 | 93.1 |
| | | 6 months | 83.6 | 88.2 | 84.7 | 95.9 | 89.7 | 91.2 | 91.0 | 92.3 | 93.2 | 93.2 |
| | all isolates | 1 year | 84.7 | 80.5 | 82.4 | 89.1 | 91.4 | 88.4 | 88.4 | 87.7 | 88.2 | 90.0 |
| | | 6 months | 80.1 | 80.9 | 84.1 | 92.4 | 89.6 | 87.8 | 89.2 | 86.5 | 90.1 | 90.0 |
| CZOP | 180 days | 1 year | 95.5 | 94.6 | 96.4 | 98.4 | 98.8 | 98.9 | 97.8 | 97.4 | 98.6 | 97.5 |
| | | 6 months | 91.8 | 97.0 | 95.9 | 100.0 | 97.2 | 98.8 | 96.7 | 97.6 | 98.4 | 96.8 |
| | 30 days | 1 year | 91.9 | 91.7 | 93.8 | 95.7 | 96.5 | 96.4 | 96.7 | 96.5 | 97.8 | 96.3 |
| | | 6 months | 87.9 | 94.7 | 91.9 | 98.2 | 93.2 | 97.4 | 95.8 | 97.3 | 97.3 | 95.5 |
| | all isolates | 1 year | 90.3 | 90.1 | 90.9 | 92.9 | 94.1 | 94.3 | 95.1 | 94.1 | 93.5 | 93.3 |
| | | 6 months | 87.7 | 92.2 | 89.4 | 95.2 | 92.2 | 95.5 | 94.6 | 93.6 | 93.4 | 93.3 |
| DRPM | 180 days | 1 year | 92.9 | 95.1 | 95.9 | 98.0 | 98.0 | 97.0 | 95.9 | 96.5 | 97.9 | 98.2 |
| | | 6 months | 92.9 | 96.2 | 95.9 | 99.3 | 96.2 | 96.5 | 95.4 | 97.6 | 98.4 | 98.1 |
| | 30 days | 1 year | 89.7 | 92.9 | 93.8 | 95.4 | 95.8 | 94.8 | 95.3 | 95.7 | 97.2 | 97.2 |
| | | 6 months | 89.7 | 94.7 | 91.9 | 97.7 | 92.3 | 95.9 | 94.6 | 96.7 | 97.9 | 96.6 |
| | all isolates | 1 year | 90.1 | 89.1 | 88.5 | 92.6 | 92.6 | 93.4 | 94.9 | 94.3 | 95.3 | 93.2 |
| | | 6 months | 90.1 | 88.2 | 88.8 | 95.2 | 88.3 | 96.2 | 93.3 | 95.1 | 95.5 | 91.1 |

(*Continued*)

**Table 6.** (Continued)

| Antimicrobials | Interval considered as duplicate isolates | Collection period | Susceptible rate (%) | | | | | | | | | |
|---|---|---|---|---|---|---|---|---|---|---|---|---|
| | | | 2014 | | 2015 | | 2016 | | 2017 | | 2018 | |
| | | | Jun | Dec | Jun | Dec | Jun | Dec | Jun | Dec | Jun | Dec |
| IPM | 180 days | 1 year | 90.5 | 87.9 | 88.7 | 88.3 | 89.7 | 90.0 | 89.6 | 92.9 | 94.7 | 90.7 |
| | | 6 months | 85.7 | 89.4 | 87.6 | 88.8 | 90.6 | 88.4 | 91.4 | 94.6 | 95.3 | 86.0 |
| | 30 days | 1 year | 85.9 | 86.1 | 85.0 | 85.8 | 88.2 | 88.0 | 89.2 | 92.2 | 93.8 | 89.1 |
| | | 6 months | 83.6 | 86.8 | 82.0 | 88.3 | 87.2 | 87.6 | 91.0 | 93.4 | 94.5 | 84.7 |
| | all isolates | 1 year | 81.6 | 79.2 | 80.5 | 83.6 | 84.0 | 84.1 | 87.3 | 90.7 | 91.7 | 83.8 |
| | | 6 months | 77.8 | 80.4 | 80.6 | 85.7 | 81.2 | 85.7 | 89.2 | 92.1 | 91.3 | 77.0* |
| MEPM | 180 days | 1 year | 93.5 | 92.0 | 94.6 | 95.2 | 94.8 | 93.0 | 92.8 | 94.6 | 96.5 | 97.1 |
| | | 6 months | 88.8 | 93.9 | 93.8 | 96.1 | 92.5 | 92.5 | 93.4 | 95.8 | 97.7 | 94.9 |
| | 30 days | 1 year | 90.3 | 89.8 | 91.5 | 92.2 | 92.7 | 90.9 | 92.2 | 93.7 | 95.4 | 95.0 |
| | | 6 months | 86.2 | 92.1 | 89.2 | 94.2 | 88.9 | 91.8 | 92.8 | 94.5 | 96.6 | 93.2 |
| | all isolates | 1 year | 89.0 | 84.3 | 85.3 | 89.5 | 88.9 | 88.0 | 90.3 | 92.3 | 92.9 | 89.4 |
| | | 6 months | 84.8 | 83.8 | 87.1 | 91.2 | 85.1 | 89.5 | 91.3 | 93.2 | 92.6 | 86.6 |
| AZT | 180 days | 1 year | 84.1 | 84.4 | 86.5 | 90.7 | 94.0 | 90.4 | 88.7 | 89.1 | 87.0 | 88.2 |
| | | 6 months | 78.6 | 88.6 | 83.5 | 95.4 | 90.6 | 89.0 | 88.7 | 88.1 | 83.6 | 91.1 |
| | 30 days | 1 year | 79.8 | 81.2 | 83.1 | 87.6 | 90.2 | 86.7 | 86.7 | 87.9 | 86.1 | 85.9 |
| | | 6 months | 75.0 | 85.5 | 79.3 | 93.0 | 86.3 | 86.1 | 87.3 | 88.0 | 82.2 | 88.6 |
| | all isolates | 1 year | 71.2 | 74.1 | 78.9 | 85.5 | 87.9 | 85.3 | 84.4 | 83.0 | 82.3 | 83.4 |
| | | 6 months | 70.2 | 77.5 | 80.6 | 88.8 | 86.4 | 84.7 | 84.2 | 82.0 | 82.6 | 84.0 |

6 month vs 1 year:

* $P<0.05$

The *E. coli* susceptibility rates to non-β-lactam antibiotics did not show regular changes due to the data collection period. The antimicrobial agent with the largest change in susceptibility rate due to differences in the data collection period was LVFX, which was calculated in December 2015 using all isolates (a 4.3% difference).

The *K. pneumoniae* susceptibility rates to β-lactam antibiotics did not show regular changes due to the data collection period. The antimicrobial agent with the largest change in susceptibility rate due to differences in the data collection period was ABPC/SBT, which was calculated in December 2014 using all isolates (a 6.5% difference).

The *K. pneumoniae* susceptibility to non-β-lactam antibiotics did not show regular changes due to the data collection period. The antimicrobial agent with the largest change in susceptibility rate due to differences in the data collection period was MINO, which was calculated in December 2014 using all isolates (a 5.7% difference).

## Influence of hospital relocation on patient dynamics

Table 7 shows the number of target patients, the number of patients who visited the pre-relocation hospital and underwent susceptibility testing, and the proportion of those patients in the target patients. All data was calculated every six months for each bacterial type.

For all bacterial species, for one year after hospital relocation patients who had consulted the pre-relocation hospital accounted for approximately 10% of all target patients. Afterwards, the proportion of patients who visited the pre-relocation hospital decreased sharply, settling under 3% in late 2018.

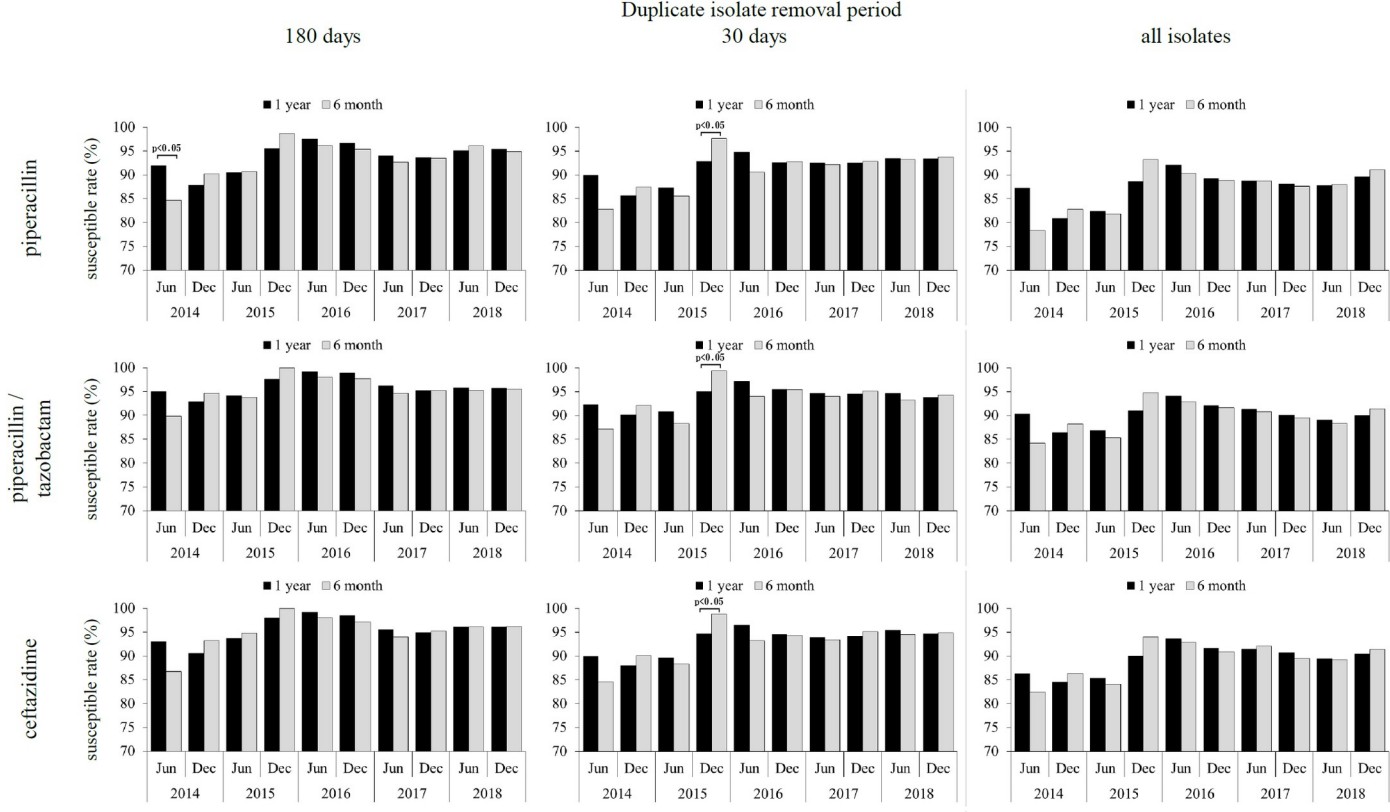

**Fig 4. Effect of data collection period on *P. aeruginosa* susceptibility rates to piperacillin, piperacillin/tazobactam, and ceftazidime.**

## Discussion

### Effect of duplicate isolate removal periods

First, this study showed that for any bacterial species, the longer the duplicate isolate removal period, the fewer isolates are available and the higher the calculated susceptibility rate. This finding is consistent with previous reports [5, 6].

When calculating the *P. aeruginosa* susceptibility rates using 1 year of data, setting the duplicate isolate removal period to 30 days reduced the available target strain by approximately 30% compared with using all the strains. When the duplicate isolate removal period was 365 days (that is, when only the first strain of the same patient was used), the available strains decreased by about 40%. In the cases of *E. coli* and *K. pneumoniae*, when the duplicate isolate removal period was 365 days, the available target strains were reduced by approximately 30% and 20%, respectively, compared with when all the strains were used.

A decrease of between 20% and 40% in target isolates resulting from setting duplicate isolate removal periods is a major problem when facilities have little antimicrobial susceptibility test data. This is also true for bacterial species with a small number of cases. To acquire adequate data in such situations, the data collection period could be extended, or data could be compiled with data from other institutions.

However, when either of these approaches are used (i.e., extending the data collection period or pooling data from other institutions), a question arises about whether the resulting antibiogram accurately reflects the true situation at the respective facility.

Next, we showed that the susceptibility rates of *P. aeruginosa* to β-lactam antibiotics were significantly higher when the duplicate isolate removal periods were set at 180 and 365 days, respectively, than when whole strains were used. The *P. aeruginosa* susceptibility rates to β-lactam antibiotics increased by 1.4%–10.7% when the duplicate isolate removal period was 365 days when compared with using all the strains.

The susceptibility rates for other combinations of bacterial species and antimicrobial agents did not show a significant increase as seen with *P. aeruginosa* to β-lactam antibiotics when increasing the duplicate isolate removal period. However, increasing the duplicate isolate removal period caused the susceptibility rate to increase within a range of several per cent.

It should be noted that susceptibility rates may change due to changes in the duplicate isolate removal period. As found in the comparison of coefficients of variation, the duplicate isolate removal period has a greater impact on the susceptibility rate in *P. aeruginosa* than in *E. coli* and *K. pneumoniae*.

Antibiograms are used to assess susceptibility rates, as an aid in selecting antimicrobials in empiric therapy, in monitoring resistance trends over time within an institution, in comparing susceptibility rates across institutions, and in tracking resistance trends [9]. Thus, the clinician should be fully aware that the antibiogram susceptibility results for any pathogen and antimicrobials, particularly *P. aeruginosa* and β-lactam antibiotics, may vary based on the duplicate isolate removal periods. Other antimicrobial agents in *P. aeruginosa*, and with respect to antibiograms of *E. coli* and *K. pneumoniae*, do not appear to be of such concern.

## Impact of the data accumulation period

In this study, it was demonstrated that when calculating the susceptibility rate every 6 months, it differs by approximately 4%–8% depending on whether the data collection period is set to 6 months or 1 year. Medical personnel using antibiograms should be aware of this fact.

As demonstrated in Table 7, the patient population changed drastically both prior to and after relocation of the Sakai City Hospital Center, which may reflect changes in the bacterial flora within and outside the hospital environment. Therefore, susceptibility trends observed in Fig 4 with a data collection period of 6 months, can be interpreted as follows.

In the first half of 2015, the *P. aeruginosa* strains circulating within and outside the pre-relocation hospital were susceptible to PIPC with values ranging from 81%–91%, and for PIPC/TAZ this ranged from 85%–95%, and for CAZ this ranged from 84%–94%. On the other hand,

**Table 7. The number of target patients every 6 months.**

| | | patients who visited the pre-relocation hospital / target patients | | | percentage of patients who visit old location (%) | | |
|---|---|---|---|---|---|---|---|
| | | *P. aeruginosa* | *E. coli* | *K. pneumoniae* | *P. aeruginosa* | *E. coli* | *K. pneumoniae* |
| 2014 | first half | - / 106 | - / 313 | - / 100 | - | - | - |
| | second half | - / 142 | - / 353 | - / 132 | - | - | - |
| 2015 | first half | - / 102 | - / 346 | - / 104 | - | - | - |
| | second half | 21 / 165 | 43 / 369 | 10 / 130 | 12.7 | 11.7 | 7.7 |
| 2016 | first half | 10 / 112 | 22 / 431 | 2 / 133 | 8.9 | 5.1 | 1.5 |
| | second half | 6 / 184 | 15 / 462 | 2 / 179 | 3.3 | 3.2 | 1.1 |
| 2017 | first half | 7 / 164 | 20 / 422 | 5 / 176 | 4.3 | 4.7 | 2.8 |
| | second half | 7 / 178 | 20 / 407 | 5 / 183 | 3.9 | 4.9 | 2.7 |
| 2018 | first half | 7 / 133 | 12 / 418 | 3 / 158 | 5.3 | 2.9 | 1.9 |
| | second half | 4 / 172 | 8 / 429 | 2 / 213 | 2.3 | 1.9 | 0.9 |

it is estimated that the *P. aeruginosa* strains circulating at the new relocation site had good susceptibility to both antimicrobials since the majority of the patients were replaced by new residents at the new site. Immediately after the move in July 2015, PIPC showed susceptibility rates of 93%–99%, while for PIPC/TAZ, this was 95%–100%, and for CAZ, this was 94%–100%. Afterwards, it is expected that hospital staff and patients from the old location site would have introduced *P. aeruginosa* from the pre-relocation hospital flora to the local bacteria at the new site, resulting in gene mixing and exchange of drug resistance genes, eventually leading to a new susceptibility profile [10, 11].

Thus, it is reasonable to suggest that the susceptibility trends observed in Fig 1 for a data collection period of 6 months reflect the changes in bacterial flora and subsequent changes due to hospital relocation. However, in the case of the 1-year data collection period, since the data from the first and second halves of 2015 were shown as data for the second half of 2015, a sharp rise in susceptibility rate was not observed. We were also unable to observe any floral changes resulting from hospital relocation and the subsequent transition.

As such, even if a significant change occurs in antimicrobial susceptibility during the analysis period, a large change in susceptibility may be obscured by calculating susceptibility rate combining data of pre- and post- change. Also, since bacteria such as *P. aeruginosa*, which do not require special nutrients for growth, easily multiply, have a variety of resistance mechanisms, and easily acquire resistance, antimicrobial susceptibility rates may change drastically over a few months. Therefore, a significant change in susceptibility rate may be missed if the data accumulation period is not as long as possible. This point should always be noted when evaluating trends in susceptibility rates of antimicrobial drugs. The CLSI guidelines recommend updating antibiograms "at least annually" [3]. However, we believe they should be updated within the shortest time period that is practically possible, taking into account restrictions such as numbers of specimen.

## Conclusions

In this study we have clearly demonstrated how antibiograms are influenced by various parameters including duplicate isolate removal periods and data collection periods. Furthermore, our data suggests that changes in antimicrobial susceptibility rates may be missed depending on the data accumulation period and timepoints used during antibiogram preparation.

In addition to the personnel involved in creating the antibiograms, the medical personnel involved in treating infectious diseases, who also interpret such antibiograms, need to be well aware of the extent to which these parameters influence antibiograms. It is also important to unify parameter values when comparing antibiograms from multiple facilities, and the parameter values used to create the antibiograms should be clearly indicated.

Ideally, medical professionals involved in treatment of infectious diseases should select antimicrobial agents based on their experience in daily clinical care and use antibiograms based on various parameter values. To do so, a system such as Chans, which allows the operator to browse and set various parameter values and conditions to rapidly calculate antimicrobial susceptibility or resistance rates, is recommended.

## Supporting information

**S1 Fig. Changes in the *P. aeruginosa* susceptibility rates to β-lactam antibiotics according to the data collection periods.**
(TIF)

**S1 Table. Difference in susceptible rate when the duplicate isolate removal periods were set at 365 days, 180 days, and 30 days, compared with when all isolates were used.** (XLSX)

## Author Contributions

**Conceptualization:** Yasutoshi Hatsuda, Syou Maki.

**Data curation:** Naonori Koizumi, Yukako Yasui, Takako Saito, Junji Mukai.

**Formal analysis:** Syou Maki, Junji Mukai.

**Investigation:** Yasutoshi Hatsuda, Sachiko Omotani, Tomoya Tachi.

**Project administration:** Sachiko Omotani.

**Supervision:** Toshihiko Ishizaka, Tomoya Tachi, Hitomi Teramachi, Michiaki Myotoku.

**Validation:** Syou Maki, Sachiko Omotani.

**Visualization:** Sachiko Omotani.

**Writing – original draft:** Yasutoshi Hatsuda.

**Writing – review & editing:** Yasutoshi Hatsuda.

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
