## [Decision Letter · Decision Letter 0]

19 Dec 2019

PONE-D-19-32239

Influence of analysis conditions for antimicrobial susceptibility test data on susceptibility rates

PLOS ONE

Dear Mr. HATSUDA,

Thank you for submitting your manuscript to PLOS ONE. After careful consideration, we feel that it has merit but does not fully meet PLOS ONE’s publication criteria as it currently stands. Therefore, we invite you to submit a revised version of the manuscript that addresses the points raised during the review process.

As noted by reviewer 1 below, the authors examined the effect of some underappreciated parameters for construction and application of antibiograms. While these results would not surprise any experts in the field, it is still worthwhile to quantify the effect properly and describe it in a manuscript. However, in this case, the authors are extrapolating a very limited data set to support the conclusion. Reviewer 1 has suggested some ideas for remediating the situation, but it will almost certainly require considerably more analysis and data. To avoid any wasted effort, the authors should evaluate carefully if the additional data is available before considering submission of a revised manuscript.

We would appreciate receiving your revised manuscript by Feb 02 2020 11:59PM. To enhance the reproducibility of your results, we recommend that if applicable you deposit your laboratory protocols in protocols.io, where a protocol can be assigned its own identifier (DOI) such that it can be cited independently in the future. For instructions see: http://journals.plos.org/plosone/s/submission-guidelines#loc-laboratory-protocols

We look forward to receiving your revised manuscript.

Kind regards,

Herman Tse

Academic Editor

PLOS ONE

Journal Requirements:

2. Thank you for including your funding statement; "The funders had no role in study design, data collection and analysis, decision to publish, or preparation of the manuscript."

Please provide an amended Funding Statement that declares *all* the funding or sources of support received during this specific study (whether external or internal to your organization) as detailed online in our guide for authors at http://journals.plos.org/plosone/s/submit-now.  

Please state what role the funders took in the study.  If any authors received a salary from any of your funders, please state which authors and which funder. If the funders had no role, please state: "The funders had no role in study design, data collection and analysis, decision to publish, or preparation of the manuscript."

Reviewers' comments:

Reviewer's Responses to Questions

**Comments to the Author**

1. Is the manuscript technically sound, and do the data support the conclusions?

Reviewer #1: Partly

2. Has the statistical analysis been performed appropriately and rigorously? 

Reviewer #1: Yes

3. Have the authors made all data underlying the findings in their manuscript fully available?

Reviewer #1: Yes

4. Is the manuscript presented in an intelligible fashion and written in standard English?

Reviewer #1: Yes

5. Review Comments to the Author

Reviewer #1: HATSUDA et al presented an interesting retrospective analysis on the effect of varying parameters like deduplication period and data accumulation period on the susceptibility rate in antibiogram for Pseudomonas aeruginosa from Jan 2014 – December 2018 in a regional core hospital tertiary emergency medical center. During the period, the medical care was also relocated on Jul 1 2015 to another location as part of a different primary medical care area. It was found that both deduplication period and data accumulation period affect the susceptibility rate, where susceptibility rate increases when the longer period of deduplication (duplicate isolate removal period) or data accumulation period is used. However, Pseudomonas aeruginosa are notorious for their ability in the upregulation of intrinsic resistance mechanisms, especially in the presence of antibiotic use. Inclusion of another organism, e.g. Enterobacteriaceae, may provide a more complete picture of how varying parameters affect susceptibility rates in antibiogram. Also, only beta-lactam antibiotics were included in this study, thus it will be interesting to see if similar changes of susceptibility rate are observed for the other classes of antibiotics, e.g. fluoroquinolones, aminoglycosides. Finally, it appears that the key message from the study is that medical professionals should be aware of the impact on cumulative antimicrobial susceptibility result when data are analyzed by different methodology, and that antibiogram result should be interpreted and used with that limitation borne in mind.

Minor issues:

Table 1 does not add much information. The total number of isolates can be put into table 2 in an additional column.

Dripenem should be spelt doripenem.

p. 13 Line 222 cefzoplan should be spelt cefozopran

6. PLOS authors have the option to publish the peer review history of their article (what does this mean?). If published, this will include your full peer review and any attached files.

Reviewer #1: No

---

## [Author Response · Author response to Decision Letter 0]

13 Feb 2020

Reviewer 1: 

Thank you for your helpful suggestions. I appreciate your time and effort spending for our manuscript. I have all of your suggestions into my revision.

1. Pseudomonas aeruginosa are notorious for their ability in the upregulation of intrinsic resistance mechanisms, especially in the presence of antibiotic use. Inclusion of another organism, e.g. Enterobacteriaceae, may provide a more complete picture of how varying parameters affect susceptibility rates in antibiogram.

→ We added Escherichia coli and Klebsiella pneumoniae to target bacteria. And we recreated data and rewrite manuscript. Although the final conclusions have not changed significantly, it has become clear that the magnitude of influence of parameters differs according to the bacterial species. 

2. Also, only beta-lactam antibiotics were included in this study, thus it will be interesting to see if similar changes of susceptibility rate are observed for the other classes of antibiotics, e.g. fluoroquinolones, aminoglycosides.

→ We added 20 more antibiotics to target antibiotics, including 2 fluoroquinolones, 3 aminoglycosides, and others. And we recreated data for 29 target antibiotics in 9 groups, and rewrite manuscript. 

Minor issues:

1. Table 1 does not add much information. The total number of isolates can be put into table 2 in an additional column.

→ As a new Table 1 was inserted to the manuscript, Table 1 and Table 2 were renamed to Table 2 and Table 3, respectively. Table 2 and Table3 had become complicated because of the increase in number of target bacterial species and target antibiotics. Therefore, we did not integrate Table 2 and Table 3 into one table.

2. Dripenem should be spelt doripenem.

→ I replaced “dripenem” with “doripenem”.

3. p. 13 Line 222 cefzoplan should be spelt cefozopran

→ I replaced “cefzopran” with “cefozopran”.

In “Revised Manuscript with Track Changes”, deleted parts are shown in strikethrough and red text, and added parts are shown in blue text. Please understand that corrections to minor grammatical or calculation errors in the manuscript that do not affect the conclusion are not shown in red or blue text.

---

## [Decision Letter · Decision Letter 1]

16 Apr 2020

PONE-D-19-32239R1

Influence of analysis conditions for antimicrobial susceptibility test data on susceptibility rates

PLOS ONE

Dear Mr. HATSUDA,

Thank you for submitting your manuscript to PLOS ONE. After careful consideration, we feel that it has merit but does not fully meet PLOS ONE’s publication criteria as it currently stands. Therefore, we invite you to submit a revised version of the manuscript that addresses the points raised during the review process.

Please address all concerns by both reviewers. In particular, the rank order may appear statistifcally or analytically significant but have doubtful meaning in real life. It will be best if the authors can come up with an alternative and more meaningful way to look at the changes in the antibiogram.

We would appreciate receiving your revised manuscript by May 31 2020 11:59PM. To enhance the reproducibility of your results, we recommend that if applicable you deposit your laboratory protocols in protocols.io, where a protocol can be assigned its own identifier (DOI) such that it can be cited independently in the future. For instructions see: http://journals.plos.org/plosone/s/submission-guidelines#loc-laboratory-protocols

We look forward to receiving your revised manuscript.

Kind regards,

Herman Tse

Academic Editor

PLOS ONE

Reviewers' comments:

Reviewer's Responses to Questions

**Comments to the Author**

1. If the authors have adequately addressed your comments raised in a previous round of review and you feel that this manuscript is now acceptable for publication, you may indicate that here to bypass the “Comments to the Author” section, enter your conflict of interest statement in the “Confidential to Editor” section, and submit your "Accept" recommendation.

Reviewer #1: (No Response)

Reviewer #2: (No Response)

2. Is the manuscript technically sound, and do the data support the conclusions?

Reviewer #1: No

Reviewer #2: Yes

3. Has the statistical analysis been performed appropriately and rigorously? 

Reviewer #1: No

Reviewer #2: N/A

4. Have the authors made all data underlying the findings in their manuscript fully available?

Reviewer #1: Yes

Reviewer #2: No

5. Is the manuscript presented in an intelligible fashion and written in standard English?

Reviewer #1: No

Reviewer #2: Yes

6. Review Comments to the Author

Reviewer #1: Hatsuda et al attempted to demonstrate the impact of data collection/analysis protocols on antibiograms for Pseudomonas aeruginosa, Escherichia coli, Klebsiella pneumoniae, through an analysis of 5 years’ worth of susceptibility data from a regional centre, including relocation to a different primary medical care area in Jul 2015. While I agree with the authors’ conclusion that antibiogram can be influenced by various parameter change and that personnel involved in creating and using the antibiogram should be aware of these limitations, the method of data analysis and presentations suffered much deficiencies in the current manuscript.

For example,

1. The statistical significance of the susceptibility rate difference of the various ‘dupulicate isolate removal periods’ should be calculated instead of the absolute change in %.

2. The ranking order of antibiotic susceptibility rate has little clinical or interpretation significance and can be omitted entirely to simplify table 3.

3. Subgroup analysis could include comparisons between Pseudomonas aeruginosa and the Enterobacteriaceae, as well as between various antibiotic groups. These further analyses could reveal that Enterobacteriaceae susceptibility data are less affected by the change in period of deduplications for beta-lactam antibiotics, and aminoglycoside group antibiotics are not significantly affected by the deduplication periods etc. Such observations could then be followed by authors’ postulation of the mechanisms, e.g. difference in antimicrobial resistance mechanisms in Pseudomonas aeruginosa and Enterobacteriaceae or for various antibiotics, and that antibiograms are affected by how the data collection parameters were set.

Reviewer #2: PONE-D-19-32239R1: Dr. Hatsuda et al report on their work in regard to varying the length of time of duplicate Gram-negative isolate removal in the creation of a single center antibiogram. The manuscript highlights important findings that persons using antibiograms need to be aware of when using antibiograms to select empiric coverage for typical Gram negative infections. The results section if very long due to the addition of the concept of ranking the order of susceptibilities. It is evident that the paper is improved from the original version and additional antimicrobials were added to the study results.

Major concern:

1) It is not clear what impact the “rank order of antimicrobial susceptibility” has to a clinician. It seems academic and, while novel, perhaps, it is not clearly based in a clinical practice. For example, no clinician choses an antibiotic based on the highest or lowest susceptibility percentage. Typically, for any given pathogen, a drug is chosen for it’s known activity, safety profile, susceptibility (as a “Yes” susceptible or “No” not susceptible) and hospital formulary/cost, availability. CLSI breakpoints, not “rank order of susceptibility,” are used clinically with the many other factors noted. Therefore, unless the authors can cite a reference where this concept is used and/or describe a clinical usefulness, this reviewer suggests that all references to “rank order of susceptibility” be placed in supplemental sections and be removed from tables in regular manuscript. Would also remove reference to them from results sub-section. Again, although the rank order of susceptibility may change based on the duplicate isolate removal times, this is not necessarily helpful information to the clinician.

Minor concerns:

Table 1: CTX is given as the abbreviation for two cephalosporins, this should be corrected before publication.

Throughout the entire manuscript (and namely in tables and figures), again, DORIPENEM and CEFOZOPRAN need spelling corrected. This is noted in Table 1 with doripenem, FIGURE 1 for both, and also in Table title text.

Line 390: This reviewer would avoid using terms such as “a few percent” as it is not customary reporting.

Discussion: Somewhere in the discussion the authors should add some reference to the fact that antibiograms are used for empiric selection of antimicrobials pending full culture results, therefore, the clinician should fully grasp that the antibiogram susceptibility results for any pathogen and antibiotic may vary, even tremendously, based on the interval of time (duration) for which duplicate isolates are removed from the calculation/percentages.

Line 461: Would remove term “settle down” and chose alternate wording.

Line 499: Last word, “required.” would replace with “recommended.”

7. PLOS authors have the option to publish the peer review history of their article (what does this mean?). If published, this will include your full peer review and any attached files.

Reviewer #1: No

Reviewer #2: No

---

## [Author Response · Author response to Decision Letter 1]

30 May 2020

Thank you for your helpful suggestions. I appreciate the time and effort you spent reviewing our manuscript. I have taken all of your suggestions into account in my revision.

Reviewer 1: 

1. The statistical significance of the susceptibility rate difference of the various ‘duplicate isolate removal periods’ should be calculated instead of the absolute change in %.

→ We used the chi-square test to assess for associations between susceptibility rate and the duplicate isolate removal periods. In addition, differences in sensibility rates between the long and short data collection periods were also tested using the chi-square test. (Line 129)

2. The ranking order of antibiotic susceptibility rate has little clinical or interpretation significance and can be omitted entirely to simplify table 3.

→ We agree with your assessment. We abandoned the idea of ranking order and have removed Table 3. 

3. Subgroup analysis could include comparisons between Pseudomonas aeruginosa and the Enterobacteriaceae, as well as between various antibiotic groups. These further analyses could reveal that Enterobacteriaceae susceptibility data are less affected by the change in period of deduplications for beta-lactam antibiotics, and aminoglycoside group antibiotics are not significantly affected by the deduplication periods etc.

→ We showed that the duplicate isolate removal periods significantly affected the susceptibility rate only in Pseudomonas aeruginosa β-lactams. We also compared differences in susceptibility rates across data collection periods using coefficients of variation to highlight differences between β-lactams and other antimicrobials in P. aeruginosa, and between P. aeruginosa and two other bacterial species.

Reviewer 2: 

1. It is not clear what impact the “rank order of antimicrobial susceptibility” has to a clinician.

→ We agree with your assessment. We abandoned the idea of ranking order of antimicrobial susceptibility and have removed Table 3.

2. Table 1: CTX is given as the abbreviation for two cephalosporins, this should be corrected before publication. Throughout the entire manuscript (and namely in tables and figures), again, DORIPENEM and CEFOZOPRAN need spelling corrected. This is noted in Table 1 with doripenem, FIGURE 1 for both, and also in Table title text.

→ I apologize for my carelessness. I have corrected Table 1 and other relevant sections in the manuscript.

3. Line 390: This reviewer would avoid using terms such as “a few percent” as it is not customary reporting..

→ I corrected the phrase "at a few percent" to "under 3%" in the relevant sections. (Line 295)

4. Discussion: Somewhere in the discussion the authors should add some reference to the fact that antibiograms are used for empiric selection of antimicrobials pending full culture results, therefore, the clinician should fully grasp that the antibiogram susceptibility results for any pathogen and antibiotic may vary, even tremendously, based on the interval of time (duration) for which duplicate isolates are removed from the calculation/percentages.

→ We have inserted the appropriate text and the reference you pointed to in the “Effect of duplicate isolate removal periods” section of the “Discussion.” (Line 331)

5. Line 461: Would remove term “settle down” and chose alternate wording.

→ I have corrected the phrase "the emergence of new susceptibility profiles, which would eventually settle down" to "eventually leading to a new susceptibility profile" in the relevant section. (Line 357)

6. Line 499: Last word, “required.” would replace with “recommended.”.

→ I have corrected the phrase "required" to "recommended" in the relevant section. (Line 390)

In “Revised Manuscript with Track Changes,” deleted parts are shown in strikethrough and red text. Text that has been added is shown in blue text. Please consider that minor grammatical corrections to the manuscript that do not affect the conclusion are not shown in red or blue text.

---

## [Editor Report · Decision Letter 2]

4 Jun 2020

PONE-D-19-32239R2

Influence of analysis conditions for antimicrobial susceptibility test data on susceptibility rates

PLOS ONE

Dear Dr. HATSUDA,

Thank you for submitting your manuscript to PLOS ONE. After careful consideration, we feel that it has merit but does not fully meet PLOS ONE’s publication criteria as it currently stands. Therefore, we invite you to submit a revised version of the manuscript that addresses the points raised during the review process.

The reviewers' comments in the last revision have been addressed satisfactorily. However, there remains a few typos/ language issues that could be resolved easily. They are as follows:

1. L86-87: Gram stain is named after its inventor, and thus "Gram" should always be spelt with a capital "G".

2. L85-86: It is not necessary to introduce the abbreviated forms of bacterial names as this is the convention in biomedical literature.

3. L92 & table 1: In biomedical literature, colistin should be classified as a polymyxin or a nonribosomal peptide.

4. L116: Consider changing to "As it is possible that isolates from the same patient among long-term in-patients may undergo ..."

5. L119: "... to prevent results from being affected ..."

6. L147: space missing from E. coli

We look forward to receiving your revised manuscript.

Kind regards,

Herman Tse

Academic Editor

PLOS ONE

---

## [Author Response · Author response to Decision Letter 2]

5 Jun 2020

Dear Dr. Herman Tse,

Thank you for your helpful suggestions. I appreciate the time and effort you spent reviewing our manuscript. We have taken all of your suggestions into account in my revision. I hope that our edits and the responses we provide below satisfactorily address all the issues and concerns you have noted.

1. L86-87: Gram stain is named after its inventor, and thus "Gram" should always be spelt with a capital "G".

→ In accordance with your advice, we have changed “gram-negative” to “Gram-negative”. (Line 83, 84)

2. L85-86: It is not necessary to introduce the abbreviated forms of bacterial names as this is the convention in biomedical literature.

→ We thank you for this comment. Accordingly, we have deleted the abbreviated forms. (Line 85, 86)

3. L92 & table 1: In biomedical literature, colistin should be classified as a polymyxin or a nonribosomal peptide.

→ We appreciate your comment on this point. Accordingly, we have changed colistin classification to “polymyxins”. (Line 92, Table 1)

4. L116: Consider changing to "As it is possible that isolates from the same patient among long-term in-patients may undergo ..."

→ Thank you for your suggestion. We have changed the sentence you pointed out as your suggestion. (Line 115)

5. L119: "... to prevent results from being affected ..."

→ Thank you for your advice. We have changed the sentence by adding “from”, which I forgot to add. (Line 118)

6. L147: space missing from E. coli

→ Thank you for your advice. The error you pointed out has been corrected. (Line 146)

In “Revised Manuscript with Track Changes,” deleted parts are shown in strikethrough and red text. Text that has been added is shown in blue text. 

Again, thank you for giving us the opportunity to strengthen our manuscript with your valuable comments. We have worked to incorporate your feedback and hope that these revisions persuade you to accept our submission.

Sincerely,

---

## [Editor Report · Decision Letter 3]

9 Jun 2020

Influence of analysis conditions for antimicrobial susceptibility test data on susceptibility rates

PONE-D-19-32239R3

Dear Dr. HATSUDA,

We’re pleased to inform you that your manuscript has been judged scientifically suitable for publication and will be formally accepted for publication once it meets all outstanding technical requirements.

Kind regards,

Herman Tse

Academic Editor

PLOS ONE
---

## [Editor Report · Acceptance letter]

12 Jun 2020

PONE-D-19-32239R3 

Influence of analysis conditions for antimicrobial susceptibility test data on susceptibility rates 

Dear Dr. HATSUDA:

I'm pleased to inform you that your manuscript has been deemed suitable for publication in PLOS ONE. Congratulations! Your manuscript is now with our production department. 

Kind regards, 

on behalf of

Dr. Herman Tse 

Academic Editor

PLOS ONE